# How to Evaluate Green Development Policy Based on the PMC Index Model: Evidence from China

**DOI:** 10.3390/ijerph20054249

**Published:** 2023-02-27

**Authors:** Xiang-Fei Ma, Yi-Fan Ruan

**Affiliations:** School of Marxism, China University of Geosciences, Wuhan 430074, China

**Keywords:** China, green development, policy evaluation, PMC index

## Abstract

Implementing green development is important to realizing a harmonious relationship between humans and nature, and has attracted the attention of governments all over the world. This paper uses the PMC (Policy Modeling Consistency) model to make a quantitative evaluation of 21 representative green development policies issued by the Chinese government. The research finds: firstly, the overall evaluation grade of green development is good and the average PMC index of China’s 21 green development policies is 6.59. Second, the evaluation of 21 green development policies can be divided into four different grades. Most grades of the 21 policies are excellent and good; the values of five first-level indicators about policy nature, policy function, content evaluation, social welfare, and policy object are high, which indicates that the 21 green development policies in this paper are relatively comprehensive and complete. Third, most green development policies are feasible. In twenty-one green development policies, there are: one perfect-grade policy, eight excellent-grade policies, ten good-grade policies, and two bad-grade policies. Fourthly, this paper analyzes the advantages and disadvantages of policies in different evaluation grades by drawing four PMC surface graphs. Finally, based on the research findings, this paper puts forward suggestions to optimize the green development policy-making of China.

## 1. Introduction

Since the reform and opening up in 1978, China’s economic development has accomplished great achievements and is ranked second in the world. However, China’s economic development has faced severe challenges, especially climate change and environmental pollution, which have become issues of great concern to the Chinese government. The frequent occurrence of natural disasters and social crises caused by environmental degradation and climate change has seriously restricted the sustainable and circular development of China’s society and economy. Although China has the second largest economy in the world, it faces daunting pressure and challenges in ecological and environmental protection. According to statistics, the health and economic loss caused by air pollution in China in 2017 was equivalent to 2.5% of that year’s GDP [1]. In recent years, in order to promote the sustainable development of China’s economy and realize a harmonious relationship between humans and nature, the Chinese government has issued a series of policies of green development to promote economic transformation and upgrade. As early as 2011, the Chinese government put forward China’s green development plan for the first time in the 12th Five-Year Plan. The 13th Five-Year Plan was issued by the Chinese central government in 2016 and since then, the concept of green development has become a main theme of development. In 2021, the Chinese government once again emphasized the strategic position of green development in China’s overall modernization drive [2]. High-quality green development policies are the basis for promoting green development. In order to promote green development, the Chinese government has issued several green development policies. For example, in terms of the green development of agriculture, the General Office of the State Council clearly proposed to promote the green development of agriculture in 2017 by innovating the system and mechanism, so as to increase farmers’ income and ensure food security. However, China’s green agricultural development still has shortcomings, especially the serious waste of water resources in the process of agricultural development. In terms of industrial green development, China’s Ministry of Industry and Information Technology has issued a notice on promoting financial support for industrial green development at the county level, which requires that the role of finance should be played in supporting industrial green transformation and strengthening green manufacturing technology innovation. However, at present, China’s finances still have deficiencies in supporting the green transformation of the industry at the county level. How is the quality of green development policies issued by the Chinese government? This is a topic of common concern. Therefore, this study aimed to evaluate the green development policies issued by the Chinese government in recent years to objectively understand the status of these policies in China.

Policy evaluation starts from the formulation of policies, evaluates the effect of policy implementation, and finds the advantages and disadvantages of policies [3]. In the past, the evaluation of public policy was mostly qualitative and its conclusion was often subjective and dependent on personal bias, lacking objectivity and scientific nature. With the development of policy evaluation research, quantitative evaluation of public policy is favored by academia because of its rigor, rationality, and objectivity [4]. Therefore, it is necessary to evaluate China’s green development policy using a quantitative method.

Green development policy plays a very important role in promoting green development. At present, there is scant literature on the quantitative evaluation of green development policies. A scientific evaluation of China’s green development policies can objectively judge the current situation and provide suggestions to adjust China’s green development policy in the future. Therefore, this paper uses the PMC Index model to evaluate China’s green development policies. The PMC (Policy Modeling Consistency) Index model can analyze internal consistency and intuitively displays the advantages and disadvantages of the various dimensions of policies through the PMC curved surface. The specific process is as follows: firstly, collect green development policy texts issued by the Chinese government and use a text mining method to deeply identify the policy’s contents and eliminate the parts that are not highly relevant. Secondly, use the content and nature of the green development policy text to design an evaluation index system of green development policies. Then, the value of the first-level evaluation index and the value of the PMC Index of each policy are calculated and a multi-input–output table is established. Finally, four PMC surface maps of green policy were drawn; the concave and convex of the surface map were observed to reflect the internal consistency of green policies and corresponding suggestions were put forward. The marginal contribution of this paper is to apply the PMC index model to assess green development policies, which is helpful to understand the status of China’s green development policies and promote the research boundary of green development policies. China is the largest developing country in the world and therefore, its green development policy evaluation can also provide inspiration for other developing countries to make better green development policies.

The second part of this paper is the literature review, which evaluates the literature on green development and policy evaluation. The third part is the research design, wherein we introduce the data source of this paper, the PMC index model, the design of the index system of green development policies, and the PMC surface graph. The fourth part is the research results, including the multi-input–output table and four PMC surface graphs of different level policies. The fifth part is the conclusion, which summarizes this paper and puts forward suggestions to optimize China’s green development policy.

## 2. Literature Review

The theme of this paper is the evaluation of China’s green development policy. Therefore, this paper reviewed the literature on green development and public policy evaluation.

Firstly, as far as the study of green development, the study about green development can trace back to the 1970s, when people realized the disadvantages of the economic development model at the cost of resource consumption. People began to think about how to coordinate the relationship between economic development and ecological environment, then the concept of green development was born [5]. In 2012, the United Nations Conference on Sustainable Development took green transformation as a key topic, and since then, people’s attention and research on green development raised to a new height. At present, there are a lot of studies about green development, most studies focus on the definition of green development, the status of green development, green development efficiency, and green transformation [6,7,8]. In particular, many scholars have evaluated the level and efficiency of green development by designing the index of green development, such as Chen et al. based on the analytic Hierarchy Process (AHP), then designed a systematic evaluation method for chemical enterprises, take Shandong Lubei enterprises as an example to evaluated the green development level and found the green development level of Shandong Lubei enterprises is constantly improve [9]. In addition, some scholars used the TOPSIS (Technique for Order Preference by Similarity to an Ideal Solution) method to measure the level of China’s industrial green development, then coupling the coordination level of industrial green development and industrial water consumption are analyzed by using the couple coordination model [10]. Luo et al. evaluated the green development efficiency of China’s Yangtze River Economic Belt from 2011 to 2019 by using the data envelopment method and the impact of the digital economy on the green development efficiency are analyzed [11]. In addition to the assessment of green development, some scholars have analyzed influencing factors of green development, including environmental regulation, technological innovation, and urbanization. For example, Feng et al.researched 165 countries and regions around the world and found the financial crisis in 2007 had a negative impact on global green development [12]. Mensah et al. studied the impact of technological innovation on green growth in 28 economies of the OECD (Organization for Economic Cooperation and Development) between 2000 and 2014 and found transport-related technologies were beneficial to green growth in the Oceania subregion, climate change technologies related to energy production and transmission were detrimental to green growth in economies of OECD [13]. In addition to green development, the academic community has analyzed green innovation, green economy, and how technology affects environmental pollution. Wasiq et al. found that green innovation have a positive impact on the economic, social, and environmental performance of SMEs in the Kingdom of Saudi Arabia [14]. Wang et al. analyzed how environmental regulations affect the development of a green economy through green technology innovation, and found that stricter environmental regulations have a negative impact on the start-up of new businesses in various industries, especially the fossil fuel industry [15]. Huang et al. analyzed the relationship between information and communication technology (ICT), renewable energy, economic complexity, human capital, financial development, and ecological footprint from 1995 to 2018, and found ICT, economic complexity, and human capital in developing countries increased pollution levels, while renewable energy significantly reduced pollution levels [16].

Secondly, in terms of the research on public policy evaluation, policy evaluation runs through the process of policy formulation and implementation, which can be divided into pre-evaluation, in-process evaluation, and post-evaluation. Whether the results of policy evaluation are scientific and accurate directly affects policy formulation in the future. In the early studies, scholars mainly used the qualitative evaluation method to evaluate public policies, but the qualitative evaluation method is no longer applicable to the evaluation and analysis of policy because of its lack of feasibility and science. At present, most scholars adopt quantitative evaluation methods to evaluate policy. Quantitative evaluation methods include a multidimensional evaluation model, multiple difference method, entropy weight method, etc. In particular, there are a lot of studies on public policy evaluation using the synthetic control method and double-difference method. For example, Castillo et. al used an integrated control method to analyze the long-term impact of tourism development policies on employment in Salta Province of Argentina and found the average annual impact of tourism on employment was 11% [17]. Brossard and Moussa tested France’s cluster policy by using the double difference method and found only the policy of “world-class” clusters had a significant positive effect on regional patent applications [18]. Cheng et al. analyzed the impact of low-carbon urban construction policies on green growth in China by using the method of differential analysis, and found that the bigger the city, the better the infrastructure, and the better the scientific and technological foundation, the more significant the positive effect of low-carbon urban construction policies on green growth [19].

Based on the literature review, it can be found that there are a lot of studies related to green development and public policy evaluation but there are few studies on green development policy evaluation by using quantitative methods. Moreover, the existing studies on policy evaluation mostly focus on the effect of policy implementation, and few start from the policy formulation and conduct the prior evaluation of policy. A good level of green development policy is the premise to promote green development. Therefore, this study decides to use the PMC index model to evaluate China’s green development policy in order to provide inspiration for the adjustment of China’s green development policy in the future.

## 3. Research Design

### 3.1. Data Sources

The purpose of this paper is to evaluate green development policies issued by the Chinese government. Therefore, the data in this study come from the green development policies issued by the Chinese central government. A total of 90 related policy documents were collected in this study. Considering the rigor and feasibility of the evaluation of policy texts, the 90 texts collected were screened and some informal policy texts such as letters and replies were eliminated. Finally, 21 green development policies were obtained (see Table 1).

### 3.2. PMC Model Construction

The traditional quantitative policy evaluation model uses quantitative methods to analyze the implemented effect of policy, while the PMC index model analyzes policy itself, not the effect of policy implementation. PMC index model is a new method to quantitatively evaluate policy proposed by Estrada, which analyzes the internal consistency of policy and emphasizes the importance of all variables in policy text being equal. In the PMC index model, all relevant variables should be included as far as possible and those weakly correlated variables should not be ignored. PMC index models can avoid the phenomenon that exists in policy evaluation methods, paying too much attention to some variables while ignoring others, because the status of all variables is equal. Therefore, all variables are regarded as binary variables in the policy evaluation process. In other words, the variable is assigned a value of 1 if it is covered by the policy, and 0 if it is not [20]. In addition, the advantages and disadvantages of policy can be found more intuitively with the help of the PMC surface [20]. Based on 21 samples of green development policies, this study evaluates green development policies by using the PMC index model, the calculation process is as follows:

Firstly, assign a value to the secondary index. Equations (1) and (2) are used to assign binary values to the second-level index of the evaluation index system of green development policy. If the secondary index of green development policy meets the criteria in Table 2, it is assigned a value of 1; otherwise, it is assigned a value of 1 (see Table 2). Secondly, the value of the first level index is calculated by using Equation (3). The value of the first-level index is greater than or equal to 0 and less than or equal to 1. Finally, the PMC index of each policy is obtained by adding the values of the first-level evaluation index of each policy. The value of the PMC index ranges from 0 to 10. Equation (4) is the calculation process.
(1)P:N[0~1]
(2)P={PR:[0~1]}
(3)$$P_{i}=\overset{n}{\underset{j=1}{\sum }}\frac{P_{\mathrm{ij}}}{T\left(P_{\mathrm{ij}}\right)}$$

In Equation (3), i = 1,2,3… n, and i is first-order index, j is second-order index.
(4)$$\mathrm{PMC}=\left[\begin{array}{c} P_{1}\left(\overset{3}{\underset{a=1}{\sum }}\frac{P_{1i}}{3}\right)+P_{2}\left(\overset{4}{\underset{b=1}{\sum }}\frac{P_{2i}}{4}\right)+P_{3}\left(\overset{3}{\underset{c=1}{\sum }}\frac{P_{3i}}{3}\right)+\\ P_{4}\left(\overset{4}{\underset{d=1}{\sum }}\frac{P_{4i}}{4}\right)+P_{5}\left(\overset{3}{\underset{e=1}{\sum }}\frac{P_{5i}}{3}\right)+P_{6}\left(\overset{2}{\underset{f=1}{\sum }}\frac{P_{6i}}{2}\right)+\\ P_{7}\left(\overset{3}{\underset{g=1}{\sum }}\frac{P_{7i}}{3}\right)+P_{8}\left(\overset{3}{\underset{h=1}{\sum }}\frac{P_{8i}}{3}\right)+P_{9}\left(\overset{3}{\underset{k=1}{\sum }}\frac{P_{9i}}{3}\right)+P_{10} \end{array}\right]$$

### 3.3. Policy Indicator System Setting

After sorting out the contents of 21 green development policy texts, the evaluation index system of green development policy in this study is established by referring to the research of Estrada, Dai et al., Khalil et al., and Laurence et al. [20,21,22,23]. There are 10 first-level variables and 28 second-level variables in the policy evaluation index system (see Table 3). Among them, X_1_~X_10_ are first-level indices, including policy nature, policy function and timeliness of policy, evaluation of policy content, the social benefit of the policy, policy incentive, policy subject, policy object, implementation guarantee of policy, and openness degree of policy.

Different first-level indicators in the policy evaluation index system represent different evaluation criteria but they are closely related to the policy. Figure 1 shows the relationship among the first-level index of this study [21]. Firstly, in the process of implementing green development policy, policy nature, policy timeliness, policy subject, and policy object affect the effectiveness of green development policy [17]. Secondly, due to the differences in the subjects’ various policies, the design of policy focuses on the policy function, policy incentives, and guarantee of policies are different, which also means the various policies are more distinctive and targeted [21]. Finally, the key point of policy evaluation is to consider whether the social benefits brought by policy implementation and whether the content of policies are comprehensive and feasible [21]. 

### 3.4. PMC Surface Drawing

The PMC curved surface map measures the consistency of the internal structure of the policy in the form of three dimensions, which can present the advantages and disadvantages of the policy [24]. Since all policies are open to access, the value of the tenth first-level index (X_10_) for measuring policy openness in all 21 policies is 1. Considering the symmetry and operability of matrix construction, the variable of X_10_ is eliminated. Finally, a 3 × 3 surface matrix is formed by nine first-level indices to draw PMC surface maps of different levels of policies. 3 × 3 surface matrix is shown in Equation (5).
(5)$$\mathrm{PMC}=\left[\begin{array}{ccc} X_{1} & X_{2} & X_{3}\\ X_{4} & X_{5} & X_{6}\\ X_{7} & X_{8} & X_{9} \end{array}\right]$$

## 4. Research Results

### 4.1. Multi-Input-Output Table Results

The multi-input-output table can reflect the value of various variables of different policies. The binary assignment method is adopted to assign secondary indicators of green development policies. If the content of the policy meets the evaluation criteria of secondary indicators, the value of the secondary indicator is assigned as 1; otherwise, it is 0 [25]. For example, the formulation content of policy P_1_ has supervision characteristics, so the value of X_1.1_ in policy P_1_ is assigned as 1. The specific value of secondary indicators of 21 green policies are shown in Table 4.

The PMC index of green development policies can be obtained according to the values of second-level indicators of various policies. Then, the evaluation levels of 21 green development policies can be determined by referring to Table 5. Based on the multi-input-output table (see Table 4) and policy evaluation grade standard (see Table 5), the PMC index of 21 green development policies are summarized and ranked, which can provide support for quantitative analysis of green development policies at different grade (see Table 6).

It can be seen from Table 6, the average PMC index of 21 green development policies is 6.59, which is a good grade on the whole, and the average values of each first-level indicator are 0.62, 0.65, 0.48, 0.70, 0.83, 0.50, 0.56, 0.70, and 0.56, respectively. Among 21 green development policies, the PMC index values of sixteen green development policies are greater than six. In addition, among twenty-one green development policies, there is one perfect grade policy, eight excellent grade policies, ten good grade policies, and two bad grade policies. The policies grade as excellent and good account for 90.5% and the PMC value of most of the twenty-one policies are higher than the average PMC value of the twenty-one policies. This shows that the twenty-one green development policies issued by Chinese government are relatively comprehensive, scientific, and systematic. Among them, the value of the PMC index of policy P_1_ is the highest, and the value is 9.17. Policy P_1_ is Opinions on Promoting Agricultural Green Development by Innovating Institutions and Mechanisms Promulgated by the General Office of the CPC Central Committee and The General Office of the State Council. Except for the policy timeliness (X_3_), the values of other eight first-level indicators in policy P_1_ are higher than the values of first-level indicators in the other twenty green development policies. The lowest PMC index value is P_7_, and the value is 4.58, referring Table 5, the grade of P_7_ is bad. P_7_ is the Notice on Promoting Financial Support for County Industrial Green Development issued by the Ministry of Industry, the Ministry of Information Technology, and the Agricultural Bank of China. Most of the first-level indicators of policy P_7_ are lower than the average of the first-level indicators of twenty-one green development policies, so the content of policy P_7_ needs to be optimized and adjusted. It can be found from Table 6 that, among the nine first-level variables of twenty-one green development policies, the value of X_5_(social benefit) is highest, and the value of X_5_ is 0.83. It can be seen from Table 6 that the PMC index values of twenty-one green development policies differ greatly. The policy with the lowest PMC index value is P_7_ (4.58), and the policy with the highest PMC index value is P_1_ (9.17). The PMC index value of P_1_ is 4.59 higher than that of P_7_ policy. In addition, it can be found from Table 6 that the first-level index values of twenty-one green development policies also differ greatly. The mean value of X_10_ is 1 and the mean value of X_3_ is 0.48, which indicate Chinese government needs to improve timeliness of green development policies in future.

It can be seen from Table 6 that the quality of green development policies issued by the Chinese government is good, which indicates the Chinese government is qualified in policy formulation to promote green development, but the actual effect of green development policy in promoting green development needs to be further analyzed.

### 4.2. Quantitative Evaluation of Green Development Policies

This study selected four green development policies with different grades for specific analysis. According to the classification of policy evaluation grade in Table 5 and considering the representativeness of policy and the differences reflected in the policy issued by different subjects, four policies are selected from perfect grade, excellent grade, good grade, and bad grade, respectively as the sample to analyze. In this study, P_1_ (perfect grade), P_3_ (excellent grade), P_17_ (good grade), and P_14_ (bad grade) are selected and the PMC surface maps of four policies are drawn by using Matlab software to analyze the advantages and disadvantages of four green development policies (see Figure 2, Figure 3, Figure 4 and Figure 5).

In Figure 2, Figure 3, Figure 4 and Figure 5, the convex part (red) on the surface indicates the corresponding index of the policy has a higher value, whereas the concave part (blue) indicates the corresponding index of the policy has a lower value. According to the values of the first-order variable of four policies (see Table 6), the PMC index value of each policy and the three-dimensional PMC curved graph, the differences, advantages, and disadvantages of each policy can be analyzed more scientifically and systematically. So as to provide inspiration for the improvement of China’s green development policy in the future. The following are the evaluation results of P_1_, P_3_, P_17_, and P_14_.

The value of the PMC index of P_1_ is 9.17, and the evaluation grade is perfect, it ranks first among twenty-one green development policies. Policy P_1_ is “Opinions on Promoting Green Agricultural Development by Innovating Institutions and Mechanisms” issued by the General Offices of the CPC Central Committee and the State Council. The purpose of policy P_1_ is to promote the green development of China’s agriculture by adhering to institutional innovation, policy innovation, technological innovation, and the supply side structural reform of agriculture. It can be seen from Table 6 and Figure 2, there is no obvious depression in Figure 2, indicating the internal consistency level of P_1_ is relatively high and there are no weak fields of P_1_. In addition to indicator X_10_, the values of the other nine first-level indicators of P_1_ are higher than the average values of the first-level indicators of the twenty-one green development policies, and only the values of policy prescription (X_3_) and policy subject variable (X_6_) are 0.67 and 0.5, indicating that P_1_ has deficiencies in policy timeliness and policy subject. In addition to the index of X_3_ and the index of X_6_ in policy P_1_, the values of the other eight first-level index are one, which indicate P_1_ is a comprehensive policy.

Policy P_3_ is assessed as an excellent grade, the value of the PMC index of P_3_ is 7.33 and the rank fourth among twenty-one green development policies. P_3_ is “Notice on Special Management Measures for Investment from the Central Budget in Major Regional Development Strategy Construction (Green Development Direction of the Yangtze River Economic Belt)” issued by the National Development and Reform Commission. In the index of P_3_, the social benefit indicator (X_5_) and policy object indicator (X_8_) are first-order variables, and the values of X5 and X8 are the highest. The minimum value of X_7_ (policy incentive) is 0.33, which is not only the lowest among the 10 first-level indicators in P_3_ but also the lowest among all first-level indicators of twenty-one green development policies. This is because the theme of the P_3_ is investment management for the green development of the Yangtze River Economic Belt, focusing on the central budget investment projects in key regions and making specific provisions on financial subsidies. However, as P_3_ is highly targeted and related to the central government’s investment in green development of the Yangtze River Economic Belt, while policy P_3_ does not involve tax incentives and talent support. It can be seen from Figure 3, the red part protrudes mostly and the curved surface fluctuates little, indicating that P_3_ is good in coordination.

The value PMC index of P_17_ is 6.08, and the evaluation grade is good, ranking fourteenth among twenty-one green development policies. P_17_ is “The Notice of “Administrative Measures for the Construction of the Pilot Support System for the Green Development of Agriculture” issued by the General Office of the Ministry of Agriculture and Rural Affairs. The task of P_17_ is to effectively strengthen the construction of the national agricultural green development pilot zone, and it performs well in the aspect of social benefits (X_5_). P_17_ is issued from the aspect of environmental and economic benefits, which is conducive to implementing the green development concepts to some extent. However, the values of X_1_, X_3_, and X_7_ of the first-level index of P_17_ are 0.33, and the value of X_2_ is 0.5, indicating P_17_ is weak in the aspects of policy nature (X_1_), policy function (X_2_), policy timeliness (X_3_) and policy incentive (X_7_). The surrounding area of Figure 4 is concave to varying degrees, which indicates the internal consistency level of P_17_ needs to be improved. It is necessary to consider how to promote the green development of agriculture through overall coordination and local optimization in the future.

The grade of P_14_ is evaluated as bad, and its value of the PMC index is 4.83, much lower than the average value of the PMC index of twenty-one green development policies. P_14_ is the “Work Program of Agricultural Mechanization to Promote Agricultural Green Development” issued by the Ministry of Agriculture. P_14_ aims to play a central role in agricultural mechanization technology and equipment to promote the green development of agriculture. P_14_ not only has deficiencies in policy nature (X_1_), and policy timeliness (X_3_), but also needs to improve in policy incentive (X_7_), policy object (X_8_), and implementation guarantee (X_9_). It can be seen from Figure 5, the convex and convex difference of Figure 5 is large, indicating that the internal consistency level of P_14_ is low. This may be related to the policy nature and the focus of P_14_, so the contents of P_14_ need to be adjusted in the future.

This part only selects four green development policies (P_1_, P_3_, P_14_, P_17_) and analyzes the advantages and disadvantages of these four policies by PMC curved graph. In the future, it is necessary to carry out a specific analysis of the remaining policies by using a PMC curve graph.

## 5. Conclusions

### 5.1. Conclusions and Suggestions

Green development policy plays an important role in promoting green development. In this paper, the PMC index model is used to quantitatively evaluate the development policies of China. The research findings are as follows: firstly, the formulation of green development policies in China is reasonable and effective on the whole, and the values of the PMC index of most green development policies are greater than six. Second, among twenty-one green development policies, nineteen policies were rated good or better, and two policies were rated bad. Third, most green development policies are feasible and cover a wide range of fields. Ministries and commissions of the Chinese government can scientifically formulate specific plans in key areas according to strategic documents issued by the Chinese central government. Fourthly, the average values of the first-level variable among twenty-one green development policies from high to low are X_5_-X_4_-X_8_-X_2_-X_1_-X_9_-X_7_-X_6_-X_3_.

Since the values of policy timeliness (X_3_), policy incentive (X_7_), and implementation guarantee (X_9_) are relatively low, this study puts forward some suggestions to improve China’s green development policies: Firstly, pay attention to long-term planning of green development policy. In formulating policies, both short-term goals and long-term plans should be taken into account, and strong coordination among different departments to ensure a long-term blueprint of green development is implemented without distortion. Secondly, to increase policy incentives, all departments should focus on improving incentive and restraint mechanisms and strong financial support for green development. Finally, improve policy guarantee of green development policy. In the process of optimizing green development policy, it is necessary to improve green development policy from the aspects of law, social supervision, and technological innovation.

### 5.2. Limitations and Prospect

In this paper, the PMC index model is adopted for the quantitative evaluation of twenty-one green development policies in China. The main limitation of this paper is that these twenty-one green development policies are issued by the central government, and the green development policies issued by the local governments of China are not evaluated. In addition, this paper mainly evaluates green development policies, there is no assessment of the actual effect of green development policy on green development. Therefore, it is necessary to expand the sources of green development policies in the future, it is not only necessary to evaluate green development policies issued by the central government, but also to evaluate green development policies issued by local governments. In addition, it is also necessary to evaluate the impact of green development policy on green development.

## Figures and Tables

**Figure 1 ijerph-20-04249-f001:**
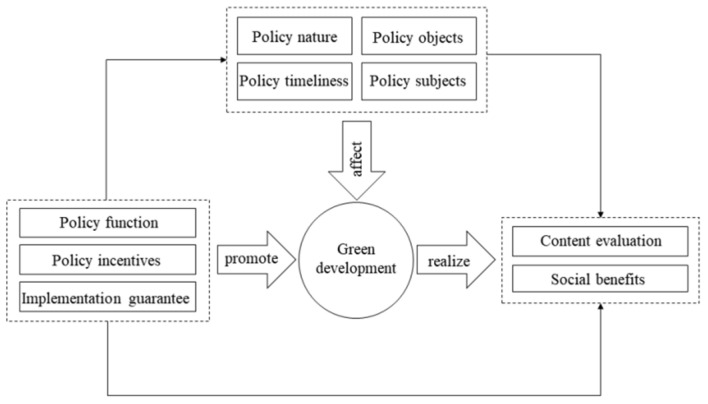
Relationship of different variables.

**Figure 2 ijerph-20-04249-f002:**
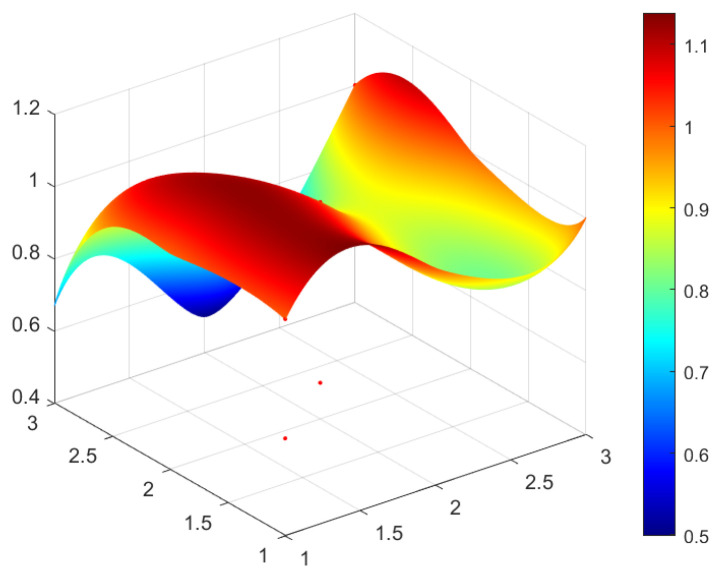
PMC surface of policy P_1_ (Perfect).

**Figure 3 ijerph-20-04249-f003:**
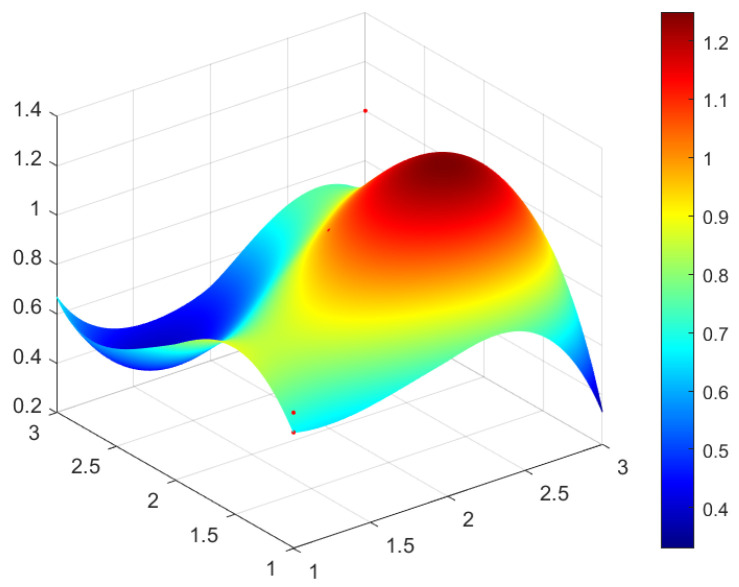
PMC Surface of policy P_3_ (Excellent).

**Figure 4 ijerph-20-04249-f004:**
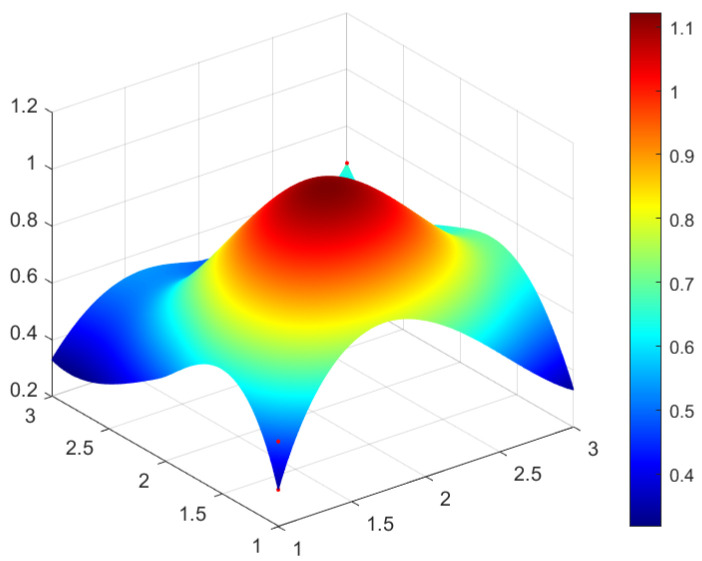
PMC surface of policy P_17_ (Fine).

**Figure 5 ijerph-20-04249-f005:**
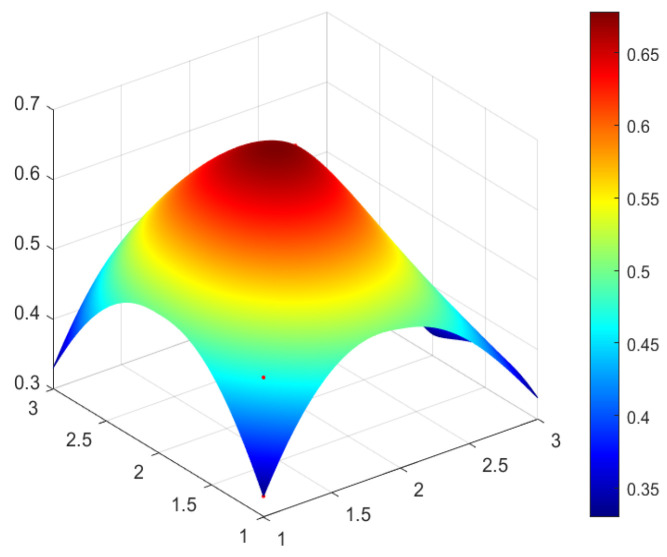
PMC surface of policy P_14_ (Undesirable).

**Table 1 ijerph-20-04249-t001:** 21 representative green development policies.

Code	Policy Name	Date Issued
P_1_	Opinions on innovating the system and mechanism to promote the green development of agriculture	10 September 2017
P_2_	Opinions on promoting green development of urban and rural construction	21 October 2021
P_3_	Measures for the administration of special investments from the central budget for the construction of major regional development strategies	9 April 2021
P_4_	A guideline on strengthening green industrial development in the Yangtze River Economic Belt	30 June 2017
P_5_	Guidelines on strengthening industry-finance cooperation to promote green industrial development	3 September 2021
P_6_	Guidelines on promoting green industrial development	20 December 2019
P_7_	Circular on promoting financial support for county industrial green development	20 November 2018
P_8_	Circular on accelerate industrial energy conservation and green development	19 March 2019
P_17_	Measures for managing the construction of a support system for green agricultural development	18 November 2019
P_18_	Guidelines on supporting green development of agriculture and rural in the Yangtze River Economic Belt	11 September 2018
P_19_	Guidelines on green development in outbound investment and cooperation	9 July 2021
P_20_	Circular on promoting the green development of e-commerce enterprises	7 January 2021
P_21_	Opinions on supporting and serving the green development of private enterprises	11 January 2019

**Table 2 ijerph-20-04249-t002:** Evaluation criteria of secondary indexes.

Number	Variable	Evaluation Content	Criteria of Evaluation
X_1_	X_1.1_ Supervision	Whether the policy has a supervisory character	If yes, it is 1; If not, it is 0
	X_1.2_ Suggestions	Whether the policy is recommended	
	X_1.3_ Forecast	Whether the policy has predictive content	
X_2_	X_2.1_ Normative guidance	Whether the policy has a normative and guiding function	
	X_2.2_ Collaboration management	Whether the policy has a collaborative management function	
	X_2.3_ Classification supervision	Whether the policy has a classification supervision function	
	X_2.4_ Overall coordination	Whether the policy has the function of overall coordination	
X_3_	X_3.1_ Short-term	Whether the policy involves short-term effects (terms < 3)	
	X_3.2_ Metaphase	Whether the policy has medium-term implications (3–5 years)	
	X_3.3_ Long-term	Whether the policy involves long-term impact (terms > 5 years)	
X_4_	X_4.1_ Detailed planning	Whether the content of the policy is detailed	
	X_4.2_ Specific objectives	Whether the policy has specific objectives	
	X_4.3_ Science goals	Whether the policy plan is scientific and reasonable	
	X_4.4_ Distinctive features	Whether the policy has customized regional characteristics	
X_5_	X_5.1_ Environmental protection	Whether the policy promotes environmental protection	
	X_5.2_ Green development	Whether policies promote green development	
	X_5.3_ Circular economy	Whether the policy promotes economic circular development	
X_6_	X_6.1_ The Central government	Whether the subject of policy is The State Council	
	X_6.2_ Ministries and Commissions of the State	Whether the main body of the policy is national ministries and commissions	
X_7_	X_7.1_ Tax concessions	Whether the policy involves tax incentives	
	X_7.2_ Financial subsidies	Whether the policy has financial subsidy measures	
	X_7.3_ Talent Support	Whether the policy has talent support	
X_8_	X_8.1_ Industry	Whether the policy targets the industry	
	X_8.2_ Enterprise	Whether the policy targets include enterprises	
	X_8.3_ Relevant departments	Whether the policy targets include relevant departments	
X_9_	X_9.1_ Legal rules	Whether the policy involves legal law	
	X_9.2_ Social supervision	Whether the policy involves social supervision	
	X_9.3_ Technological innovation	Whether the policy involves technological innovation	
X_10_	Whether the policy is open and transparent;		

**Table 3 ijerph-20-04249-t003:** Evaluation index system of green development policy.

Number	First-Level Index	Number	Second-Level Index	Number	Second-Level Variables
X_1_	Policy nature	X_1.1_	Supervise	X_1.2_	Suggest
		X_1.3_	Forecast		
X_2_	Policy function	X_2.1_	Normative guidance	X_2.2_	Cooperative management
		X_2.3_	Classification supervision	X_2.4_	Overall coordination
X_3_	Policy timeliness	X_3.1_	short-term	X_3.2_	metaphase
		X_3.3_	long-term		
X_4_	Content evaluation	X_4.1_	Detailed planning	X_4.2_	Specific goal
		X_4.3_	Scientific program	X_4.4_	Distinctive feature
X_5_	Social benefits	X_5.1_	Environmental protection	X_5.2_	Green development
		X_5.3_	Circular economy		
X_6_	Policy subjects	X_6.1_	Central government	X_6.2_	National ministries and commissions
X_7_	Policy incentive	X_7.1_	Tax incentives	X_7.2_	Financial subsidy
		X_7.3_	Talent support		
X_8_	Policy object	X_8.1_	industry	X_8.2_	Enterprise
		X_8.3_	Relevant department		
X_9_	Implementation guarantee	X_9.1_	Rule of law	X_9.2_	Social supervision
X_10_	Policy disclosure	X_9.3_	Technological innovation		

**Table 4 ijerph-20-04249-t004:** Multi-input-output table.

	**X_1_**			**X_2_**				**X_3_**		
	**X_1.1_**	**X_1.2_**	**X_1.3_**	**X_2.1_**	**X_2.2_**	**X_2.3_**	**X_2.4_**	**X_3.1_**	**X_3.2_**	**X_3.3_**
P_1_	1	1	1	1	1	1	1	1	1	0
P_2_	1	1	1	1	0	1	1	1	1	1
P_3_	1	1	0	1	1	0	1	1	1	0
P_4_	0	1	0	1	1	0	1	1	0	0
P_20_	1	0	1	1	1	0	1	1	1	0
P_21_	1	0	0	1	1	1	0	1	0	0
	**X_7_**			**X_8_**				**X_9_**		
	**X_7.1_**	**X_7.2_**	**X_7.3_**	**X_8.1_**	**X_8.2_**	**X_8.3_**		**X_9.1_**	**X_9.2_**	**X_9.3_**
P_1_	1	1	1	1	1	1		1	1	1
P_2_	1	1	1	1	1	1		1	1	0
P_3_	1	0	0	1	1	1		1	0	1
P_4_	1	0	0	1	1	0		0	1	1
P_20_	1	0	1	1	1	1		0	1	1
P_21_	0	1	0	1	1	1		1	1	0

**Table 5 ijerph-20-04249-t005:** Classification of policy evaluation levels.

Index	0 ≤ PMC < 5	5 ≤ PMC < 7	7 ≤ PMC < 9	9 ≤ PMC < 10
Evaluation (Grade)	Bad policy (IV)	Good policy (III)	Excellent policy (II)	Perfect policy (I)

**Table 6 ijerph-20-04249-t006:** Policy PMC index and type.

	X1	X2	X3	X4	X5	X6	X7	X8	X9	X10	PMC Index	Type	Rank
P_1_	1.00	1.00	0.67	1.00	1.00	0.50	1.00	1.00	1.00	1.00	9.17	Perfect	1
P_2_	1.00	0.75	1.00	1.00	0.67	0.50	1.00	1.00	0.67	1.00	8.58	Excellent	2
P_3_	0.67	0.75	0.67	0.75	1.00	0.50	0.33	1.00	0.67	1.00	7.33	Excellent	4
P_4_	0.33	0.75	0.33	0.75	0.67	0.50	0.33	0.67	0.67	1.00	6.00	Good	16
P_5_	0.67	0.50	0.33	1.00	1.00	0.50	0.33	0.67	0.67	1.00	6.67	Good	11
P_6_	0.33	0.75	0.33	0.50	0.67	0.50	0.33	0.67	0.33	1.00	5.42	Good	17
P_7_	0.33	0.50	0.33	0.25	0.67	0.50	0.33	0.33	0.33	1.00	4.58	Bad	21
P_8_	0.67	0.50	0.33	0.75	0.67	0.50	0.33	0.33	0.33	1.00	5.42	Good	17
P_9_	0.67	0.50	0.33	0.50	0.67	0.50	0.67	0.67	0.67	1.00	6.17	Good	13
P_10_	0.67	0.50	0.67	0.50	1.00	0.50	0.67	1.00	0.33	1.00	6.83	Good	10
P_11_	1.00	0.50	0.33	0.75	1.00	0.50	0.67	0.67	0.67	1.00	7.08	Excellent	7
P_12_	0.67	0.75	0.67	1.00	0.67	0.50	0.67	0.67	0.67	1.00	7.25	Excellent	6
P_13_	0.67	0.75	0.33	0.75	1.00	0.50	0.67	0.67	0.67	1.00	7.00	Excellent	9
P_14_	0.33	0.50	0.33	0.50	0.67	0.50	0.33	0.33	0.33	1.00	4.83	Bad	20
P_15_	0.33	0.75	0.33	0.50	0.67	0.50	0.33	0.67	0.33	1.00	5.42	Good	17
P_16_	1.00	0.75	0.67	0.75	1.00	0.50	0.67	0.67	0.33	1.00	7.33	Excellent	4
P_17_	0.33	0.50	0.33	0.75	1.00	0.50	0.33	0.67	0.67	1.00	6.08	Good	14
P_18_	0.67	0.75	0.67	0.50	0.67	0.50	0.67	0.33	0.33	1.00	6.08	Good	14
P_19_	0.67	0.50	0.33	0.75	1.00	0.50	1.00	0.67	0.67	1.00	7.08	Excellent	7
P_20_	0.67	0.75	0.67	0.75	1.00	0.50	0.67	1.00	0.67	1.00	7.67	Excellent	3
P_21_	0.33	0.75	0.33	0.75	0.67	0.50	0.33	1.00	0.67	1.00	6.33	Good	12
Mean value	0.62	0.65	0.48	0.70	0.83	0.50	0.56	0.70	0.56	1.00	6.59	Good	-

## Data Availability

Not applicable.

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
