# Peer review of "How to Evaluate Green Development Policy Based on the PMC Index Model: Evidence from China"

_ijerph, 2023, doi:10.3390/ijerph20054249_

Round 1
Reviewer 1 Report
The manuscript explores the theme of environmental challenges in China, with specific reference to green development. The authors study how to measure this development and propose the adoption of the PMC Index Model. 21 representative green development policies issued by the Chinese government are considered to define the Model. The methodology adopted is interesting, but perhaps it could also be supplemented by more qualitative reflections.
The introduction (Lines 24-75) is well written but could be improved by better establishing the originality of the research aim, providing more details on the policies implemented in China to date. The related bibliography should also be implemented.
There are important typos and spacing issues between words, please fix them in the revision. For example see Table 1, after Line 145: "Police name"; see Lines 403, 409, 412, 413 and so on. In general, a re-reading is suggested (with particular regard to References-Lines 374-418) in order to perfectly comply with editorial rules.
In table 3 (Evaluation criteria of secondary indexes) I suggest to avoid repeating "If yes, it is 1; If not, it is 0" 29 times. It could be written once in the text.
Reviewer 2 Report
1. The methodology and results are satisfactory, but more information is required. Explain the PMC (Policy Modeling Consistency) model in greater depth. Why did you use this model and how is it superior to others?
2. The introduction is too concise. The introduction should explain what questions are you responding to and how they are beneficial to the country and the peoples? What knowledge gaps does that fill in the research? Is it relevant to current scenarios in any way?
3. The literature review section is also short; authors can strengthen it by including some recent literature, as suggested below:
https://www.mdpi.com/2071-1050/15/3/2447,
https://www.sciencedirect.com/science/article/abs/pii/S0313592622001722,
https://www.sciencedirect.com/science/article/abs/pii/S0160791X21003286,
4. There are some typos, such as in table 5, another language is used instead of English.
5. The conclusions and abstract are nearly identical. Please redo both.
6. What are the study's limitations and recommendations? The authors must address these explicitly in the conclusion section.
7. The paper's framework is ok, but still, it requires considerable revision.
Round 2
Reviewer 2 Report
The authors did an excellent job revising their manuscript. Because all of my comments have been addressed properly, the manuscript can be considered for publication.